# Genome-Wide Screening of AP2 Transcription Factors Involving in Fruit Color and Aroma Regulation of Cultivated Strawberry

**DOI:** 10.3390/genes12040530

**Published:** 2021-04-05

**Authors:** Lixia Sheng, Cong Ma, Yue Chen, Hongsheng Gao, Jianwen Wang

**Affiliations:** College of Horticulture and Plant Protection, Yangzhou University, Yangzhou 225009, China; lxsheng@yzu.edu.cn (L.S.); nlsgwjw@foxmail.com (C.M.); 13390780572@189.cn (Y.C.); hsgao@yzu.edu.cn (H.G.)

**Keywords:** *Fragaria × ananassa* Duch, AP2 transcription factor, fruit color, fruit aroma

## Abstract

*Fragaria × ananassa* Duch, which among the youngest fruit crops, comprises many popular cultivars that are famous for their favored color and aroma. The regulation roles of AP2/ERF (APETALA2/ethylene-responsive element-binding factor) transcription factors in fruit flavor and color regulation have been studied in several fruit crops. The AP2 family of strawberry, which was ignored in recent AP2/ERF identification studies, was explored in this study. A total of 64 FaAP2 (*Fragaria × ananassa* AP2) transcription factors belonging to the euAP2, euANT (AINTEGUMENTA), and baselANT groups were identified with canonical insertion motifs in two AP2 domains. The motif identification illustrated that motifs 1, 5, and 2 indicated a corresponding AP2 domain repeat 1 with a linker region, and motifs 6, 4, 3 indicated a corresponding AP2 domain repeat 2, all of which were highly conserved. By synteny analysis, *FaAP2* paralogs were identified in each sub-genome, and *FaAP2* gene duplication and loss explained the unequal AP2 loci of sub-genomes. The expression profile in three cultivars indicated that six *FaAP2* paralogs—four *WRI* (WRINKLED) gene homologs and two *AP2* gene homologs—were candidate regulators of red fruit color and/or special fruit aroma. All these finds provide a basis for further investigations into role of AP2 in fruit color and aroma and would be helpful in the targeted selection of strawberry fruit quality to improve breeding.

## 1. Introduction

The cultivated strawberry (*Fragaria × ananassa* Duch.) with favored aromas and colors is one of the most popular soft fruits worldwide. Its earliest cultivar with complicated allo-octoploid (2n = 8× = 56) was one of the youngest fruit species (approximately 300 years ago) that originated from spontaneous hybrids of two wild octoploid species *Fragaria chiloensis* subsp. *chiloensis* and *Fragaria virginiana* subsp. *virginiana* [1,2,3]. Though the chromosome-scale genomes of octoploid *F. × ananassa* ‘Camarosa’ and several diploid ancestor candidates have been assembled, the origin and evolution of its genomes have been debatable [4,5,6,7]. Two supposed sub-genome donors (*Fragaria vesca* subsp. *bracteate* and *Fragaria iinumae*) were accepted by most studies, while the other two donors were confusing and the newly recognized donors (*Fragaria viridis* and *Fragaria nipponica*) were denied by recent studies [5,7]. Furthermore, the genomic backgrounds of favored strawberry cultivars were found to be more complicated due to the fact that octoploid and diploid species have been used as parents of many commercial varieties [8,9]. For example, a decaploid cultivar *F. × ananassa* × *F. nilgerrensis* ‘Tokun’ (TK), which was popular for its special flavor being rich in peachy aroma, was raised from crossing between of *F. × ananassa* ‘Toyonoka’ and *F. nilgerrensis* [10]. More generally, many cultivars have been crosses of two octoploid subspecies like the most popular cultivar ‘Benihoppe’ (BH) and the white-fruit cultivar ‘Snow White’ (SW). Though cultivated strawberry cultivars with important fruit quality traits are excellent material for detecting the genetic locus of interesting traits like fruit colors and aromas, homologs need to be carefully examined for more introgression or recombination events because their genomes are mosaics of phylogenetically diverse progenitors [5,8]. 

The APETALA2 (AP2) transcription factor (TF) family, which contains more than one AP2 domain comprising 60–70 amino acid residues (aa), belongs to the AP2/ethylene-responsive element-binding factor (AP2/ERF) superfamily including the ERF TFs containing one AP2 domain and RAV TFs containing one additional B3 domain [11]. The angiosperm AP2 family is divided to two lineages with three groups based on phylogeny of domain evolution. Lineages of AINTEGUMENTA (ANT) and euAP2 are identified by the presence/absence of a 10-aa insertion in AP2 repeated units 1 (R1) and a 1-aa insertion in AP2 repeated units 2 (R2) [12,13]. The euANT and baselANT groups were further identified by the presence/absence of three motifs in the pre-domain region of the ANT lineage [12,13]. For the ANT lineage, AP2 TFs of the baselANT group, which is also named *WRINKLED* gene (WRI), are master regulators of oil biosynthesis because they provide lipid biosynthetic precursors [14,15]. Meanwhile, AP2 TFs of the euANT group are master regulators of the establishment and maintenance of meristems in almost all organs [16,17]. For the euAP2 lineage, the conserved miRNA172-AP2 regulation module plays key role in the morphogenesis and growth of floral organs and fruits [16]. Though the fruit flavor regulation role of AP2/ERF TFs is being elucidated in apples, sweet oranges, and grapes by metabolism regulation including anthocyanin and volatile compounds, the potential roles of AP2 TFs in strawberry are ambiguous and several AP2/ERF genome-wide identification studies only paid attention to the tissue specificity and abiotic stress response of the ERF family [18,19,20].

This study aimed to screen AP2 candidate TFs of *F. × ananassa* involved in the fruit color and/or aromas. The prediction and grouping of FaAP2 (AP2 TFs of *F. × ananassa*) proteins based on phylogeny, exon–intron structure analysis, and motif analysis were performed to identify all *FaAP2* genes. Then, comparative mapping among sub-genomes and between sub-genome donors *F. vesca* and *F. iinumae* were performed to identify homologs. Expression profiles are TK, BH, and SW were built to detect cultivar-specific expressed FaAP2 genes. The candidate TFs could enable the selection of appropriate genes for the further functional characterization of fruit color and aromas of cultivated strawberry, and our study would be helpful in understanding roles of AP2 TFs in fruit quality regulation.

## 2. Materials and Methods

### 2.1. Identification of FaAP2 Family

The *Fragaria × ananassa* genome was obtained from Genome Database for Rosaceae (https://www.rosaceae.org/, accessed on 1 March 2021). Using a hidden Markov model (HMM) search of the AP2 domain (PF00847, http://Pfam.sanger.ac.uk/, accessed on 1 March 2021) using HMMER3.3.2 [21] with a cutoff threshold value of E-5, AP2 domain-containing proteins were screened from the proteome. Then, ambiguous or incomplete AP2 domains were rejected by SMART (http://smart.embl-heidelberg.de/, accessed on 1 March 2021) prediction under a normal model and manual inspection, and only candidate proteins with at least one AP2 domain were collected. To distinguish the AP2 TF family of *F. × ananassa*, the AP2/ERF superfamily (AP2, ERF, and RAV TF families) of *Arabidopsis thaliana* and corresponding assignment rules were obtained from PlantTFDB (http://planttfdb.cbi.pku.edu.cn/, accessed on 1 March 2021). A neighbor-join (NJ) tree of candidates and the AtAP2/ERF (AP2/ERF TFs of *A. thaliana*) superfamily were built by MEGA 7 with 1000 bootstrapping replications [22]. According to topological structure, the AtAP2 homologous candidates were identified as FaAP2 TFs.

### 2.2. Lineage Analyses of FaAP2 Proteins

Based on the alignment of full-length amino acid sequences of FaAP2s and AtAP2s done using MAFFT with default parameters (https://mafft.cbrc.jp/alignment/server/, accessed on 1 March 2021), a maximum-likelihood phylogeny with the optimal model (JTT, G, and I) of amino acid substitution recommended by ModelGenerator (http://mcinerneylab.com/software/modelgenerator/, accessed on 1 March 2021) was constructed. The reliabilities of the maximum-likelihood (ML) phylogenies were tested with 100 bootstrapping replicates. Representative domains of each lineage were aligned by MUSCLE with default parameters [23].

### 2.3. Gene Structure, Domain, and Motif Analysis of FaAP2 Family

The domains and motifs were predicted by SMART under a normal model and the MEME 5.3.3 web tools (https://meme-suite.org/meme/, accessed on 1 March 2021) under a classic model, respectively. The gene structure, domains, and motifs were illustrated by the Gene structure view tool of TBtools 1.086 [24].

### 2.4. Synteny Analysis of FaAP2 Gene

An intra-species synteny analysis was conducted by a reciprocal BLASTP search for potential homologous gene pairs (*E* < 10^−5^, top five matches) in the *F. × ananassa* genome. The homologous pairs were used for MCScanX [25] to identify the syntenic regions among the sub-genomes. All the syntenic regions and homologous pairs were illustrated by Advanced Circos of TBtools 1.086 [24]. For inter-species synteny analysis, AP2 TFs of *F. vesca* and *F. iinumae* (gene ID in Appendix A) were identified with same method of AP2 TFs of *F. × ananassa*. Then, orthologous pairs in the ‘Fve sub-genome versus *F. vesca* genome’ and the ‘Fii sub-genome versus *F. iinumae* genome’ were identified by OrthoFinder with default parameters [26]. Finally, Circos illustrations were visualized as above.

### 2.5. Expression Analysis in Three Cultivars

Three cultivated strawberries—‘Benihoppe’ (BH), ‘Tokun’ (TK), and ‘Snow White’ (SW)—were planted in greenhouses of Yangzhou University (32.391° N, 119.419° E), Yangzhou, China. All plants were planted in 10 cm diameter (Φ 10 cm) pots at 20 ± 1 °C in a 16-h light/8-h dark photoperiod. The RNA of mature fruits of three varieties (3 replications for each) was extracted by the MiniBEST Universal RNA Extraction Kit (TaKaRa, Japan) according to the manufacturer’s instructions. The transcriptome sequencing of 9 samples was performed on an Illumina NovaSeq 6000 system. After acquiring clean reads via low-quality read filtration, read counts were calculated by mapping reads to the genome using HISAT 2.2.1 (http://daehwankimlab.github.io/hisat2/, accessed on 1 March 2021). TPM (transcripts per kilobase of exon model per million mapped reads) were calculated based on read counts normalization using a self-built R script. The expression profile of FaAP2 was selected from the TPM with mean values that were greater than threshold value of 1 (Appendix A). The differentially expressed analysis was based on clean reads using the R package DESeq2 (http://www.bioconductor.org/packages/release/bioc/html/DESeq2.html, accessed on 1 March 2021) by the formula abs[log2(fold-change)] > 1, and then the fold-changes of FaAP2 were fetched. 

## 3. Results

### 3.1. Identification and Lineage Division of FaAP2 Family

From the proteome of cultivated strawberry, 498 candidates of the AP2/ERF superfamily were predicted, and 64 FaAP2 TFs were distinguished from 23 FaRAV (RAV TFs of *F. × ananassa*) TFs and 411 FaERF TFs based on the corresponding topological lineages of an NJ tree with the reference of the *A. thaliana* AP2/ERF superfamily (Appendix A and Appendix A). Furthermore, topological branches of ML tree grouped the 64 FaAP2 TFs into the euAP2 and ANT lineages, including the euAP2, euANT, and baselANT groups (Figure 1). According to the homology pairs of the FaAP2 and AtAP2 TFs (Figure 1), as well as their sub-genome locations (Appendix A), the symbols of *FaAP2* gene were named ‘FaAP2 homology’–‘sub-genomes i, ii, iii, or iv’–‘number’ (if more than one *FaAP2* genes were homologous to same *AtAP2*). Several *FaAP2* genes with only one AP2 domain were further recognized based on known canonical motifs. For example, FaWRI2iv without R2 and FaWRI3/4iv4 without R1 were recognized as part of the ANT lineage by the presence of the R1 insertion motif and the R2 insertion motif (red boxes of Figure 1), respectively. Additionally, FaSMZ/SNZiii2 (SMZ/SNZ TFs of *F. × ananassa*, lack of conserved R2) was recognized as part of the euAP2 group due to its conserved microRNA (miRNA) response element (MRE) of miR172 (Appendix A). Gene symbols, gene IDs, sub-genomes, chromosomes, and AP2-domain repeats are listed in Appendix A.

### 3.2. The Gene Structures, Protein Domains, and Motifs of FaAP2 Family

After comparing the 10 most enriched motifs (Appendix A) of the MEME prediction with AP2 repeat locations, motifs 1, 5, and 2 (found to correspond to the R1 region, including linker regions) and motifs 6, 4, and 3 (found to correspond to the R2 region) were identified in nearly all FaAP2 proteins (Figure 2A). For R2, the ANT lineage had more conserved 6–4–3 motif units (except for four WRI2L homologs) than the euAP2 lineage, whose motif 6 or motif 4–3 units were lost (incomplete domain). For R1, the ANT lineage had a motif the 9–1–5 unit, except for the FaWRI1/FaWRI2 cluster that lost motif 9, whereas the euAP2 lineage only had motif 1 in most members. With respect to gene structure, all *FaAP2* genes were found to possess introns, and the average number of introns per the *FaAP2* gene of the euAP2 lineage (10) was found to be a little more than that of the ANT lineage (7–9) (Figure 2B). The short pre-domain of the FaWRI proteins encoded by the first short exon of each *FaWRI* gene indicated that the baselANT group lost a supposed exon in accordance with the second exons of the euANT group. Meanwhile, the last long exon encoding pre-domain may have split into three short exons in each *FaAP2* gene of the euAP2 group.

### 3.3. AP2 Loci Duplications and Losses in Subgenome by Synteny Analysis 

A chromosomal location analysis demonstrated that except for the seventh chromosome (Chr7) on each of the four sub-genomes, *FaAP2* genes were interspersed among the whole genome (the other 24 chromosomes) (Figure 3A). An intra-species synteny analysis indicated that each group of the paralogs (mostly four *FaAP2* genes) identified in the above phylogeny (Figure 1) was located in each collinear chromosome (Chr1–4, 6, and 7; the color lines of Figure 3A) of four synteny sub-genomes of allopolyploid (the grey lines of Figure 3A). The different number of *FaAP2* genes on the corresponding sub-genomes (18, 17, 15, and 14 on Fve, Fvi, Fii, and Fni, respectively) came from the TOEL-1(TOE like gene), TOEL-2, WRIL-1 (WRI like gene), AIL2 (ANT like gene), AIL6/7, SMZ/SNZ-1, and SMZ/SNZ-2 loci (Figure 3D and Appendix A). To detect *FaAP2* evolution in different sub-genomes, the duplication and loss events of AP2 loci were identified by inter-species synteny analysis with *F. vesca* (*FveAP2*) and *F. iinumae* (*FiiAP2*); 2/3, 0/3, 3/1, and 1/0 duplication/loss events happened in the Fii, Fni, Fve, and Fvi sub-genomes, respectively, when 16 AP2 genes of the *F. vesca* genome were used as reference loci (Figure 3B,D). Additionally, orthologous pairs of the ML phylogeny of the FveAP2 and FaAP2 proteins (Appendix A) were in full accordance with the syntenic AP2 pairs, except for FveAIL6/7 in the unassembled scaffold (homolog of FaAIL6/7 in Chr1). In the same way, 4/3, 4/2, 2/4, and 1/2 duplication/loss events were identified in Fii, Fni, Fve, and Fvi sub-genomes, respectively, when the *F. iinumae* genome was used as a reference (Figure 3D and Appendix A). However, there were several nonconformities between the syntenic pairs and the phylogenetic pairs. Three linked loci pairs (FaAIL1i–FiiAIL2, FaWRI3/4i1–FiiWRI3/4i1-1 and FaAIL1i–FiiAIL6/7; see blue lines of Figure 3C) were not supported by the phylogeny of the FiiAP2 TFs (Appendix A), and the FaWRI3/4i–FiiWRI3/4-2 pair (grey line of Figure 3C) was only identified in phylogeny. Comparisons of the loss and duplication events between the two FiiAP2 reference and the FveAP2 reference indicated that more duplication and less loss of AP2 loci happened in the Fve sub-genome with most *FaAP2* genes (Figure 3D).

### 3.4. Expression Profiles of FaAP2 in Three Cultivars

The *F. × ananassa* BH, SW, and TK cultivars are famous commercial varieties with excellent fruit quality. Regarding aroma, our previous aroma type study (unpublished, on submission) grouped BH, TK, and SW as fruity, peachy, and floral aromas, respectively. Regarding color, red BH, pink TK, and white SW are significantly different. Considering the extreme differences of fruit aroma and color among the three cultivars, *AP2* gene expression profiles were built to identify potential TFs involved in fruit color or/and aroma, and 23 unexpressed FaAP2s TFs and 29 low abundant FaAP2s were excluded. The expression profiles (Figure 4) of five *FaWRIs* (FaWRI1i–iv and FaWRI2ii), four *FaAP2s* (FaAP2i–iv), and three *FaTOELs* (ii1, iv1, and iii2) were clustered into three classes of expression bias. *FaWRI2ii*, *FaAP2iii*, and *FaTOELii1* of cluster I, which were more abundant in SW than in BH and TK, could be involved in the secondary metabolism of the aroma or/and color of SW. Similarly, highly expressed *FaAP2* genes of clusters II and III could be potential regulators of their corresponding aroma (fruity aroma and peachy aroma, respectively) and/or red color. Furthermore, three, two, and one differentially expressed genes were identified in clusters III, I, and II, respectively, when using SW as a control group, and these *FaAP2* genes could be potential regulators of fruit aroma and color. For example, FaWRI1i–iii, which are upregulated in both BH and TK, could play roles in their red color or different aromas compared with SW. FaAP2i, which is upregulated in TK, could be regulator of the unique peachy aroma of TK, while FaWRI2ii and FaAP2iii could be regulator of the floral aroma or/and white color, respectively, of SW.

## 4. Discussion

### 4.1. The Conservation and Envolution of FaAP2

The R1-linker region–R2 region was more conserved than the pre-domain and post-domain regions in the AP2 TF family of land plants, and the conserved region was essential for AP2 function [13,17]. Except for three R1-missing and eight R2-missing FaAP2 TFs, all the other AP2 TFs, including those in the euANT group, had the full conserved region. Anyway, these R1 or R2 missing FaAP2 TFs were homologs of well-confirmed AtAP2 TFs (WRI, SMZ, SNZ, and TOE). The insertion motif ‘NSC[K/R][K/R]EGQ[T/S]’ of R1 was also presented in most FaAP2 TFs of the ANT lineage, and the relatively long pre-domain presented in the euANT group. Unlike the conserved MRE site located in the post-domain of *A. thaliana* homologs [27,28], some FaAP2 TFs of the euAP2 group were absent the miR172 recognition site, thus indicating the high variability of the post-domain (Appendix A). Furthermore, the motif analysis demonstrated that the protein structure of FaAP2s is most conserved around the linker region, and divergence between the two lineages was significant in motif 5, which overlapped with 10 aa insertion of R1. Overall, FaAP2 TFs were conserved in the AP2 domains, and the euANT group performed the highest conservation in three groups.

Though there were 64 FaAP2 genes (exactly four times of the 16 FveAP2 genes), the number of AP2 loci were not equal and decreased in order of the Fve, Fvi, Fii, and Fni sub-genomes [2]. The different gene numbers could have resulted from the differential evolution of the sub-genomes or gene loci [5,29]. The absence or presence of putative ancestral AP2 loci among the four sub-genomes (Figure 3D) indicated gene loss or duplication events that resulted in the loci assignment difference [3,4]. Tandem gene duplication is a major way to generate gene clusters [30]. Several replicated loci like SMZ/SNZ-2 (red gene labels in Figure 3B) could have resulted from tandem duplication of a paralogous gene in an adjacent location within 100 KB (e.g., SMZ/SNZ-1), and duplicate AP2 genes were usually clustered in a high confident lineage (like FaSMZ/SNAiii1 and FaSMZ/SNAiii2 in the Appendix A phylogeny). Other gene expansion/contraction processes like genome duplication, segmental duplication, ectopic recombination, and selective loss [30,31] need further analysis with more comprehensive data. 

### 4.2. Potantical Regulation Roles of WRI and AP2 Involved in Fruit Aroma or Color

Only FaAP2 TFs of the AP2, WRIL-1, and WRI loci were found to be expressed in mature strawberry fruit. There were no FaAP2 TFs of the euANT group expressed in the fruit, which explained the tissue specificity of ANT/AIL in the lateral meristem [32,33].

WRI TFs regulate oil biosynthesis by providing precursors (acyl chain and glycerol backbones) for various lipid biosynthetic pathways in both seed oil and flower or fruit surface waxes [14,15,34]. However, the roles of WRI TFs in fruit aroma have not been studied. In our study, three FaAP2 genes of WRI1 seemed to be fruit aroma regulator candidates since esters are important volatility compounds of fruits. The role of AP2 in the specification of floral organ identity, establishment of floral meristem, and development of the ovule and seed coat have been well-studied [16]. Recent studies of the miR172-AP2 module indicated miR172 plays roles in promoting rice quality (increasing amylose content) [35], peach aroma weakening during cold storage [36], and tomato color enhancement during fruit maturation [19] via the suppression of AP2 and/or paralogs. Little is known about AP2’s precise role involved in the process of fruit color and aroma regulation. Whether the FaAP2iii candidate with MRE is regulated by miR172 and involved in color or aromas needs to be further studied. Additionally, other FaAP2 TFs that are expressed relatively highly like FaTOELii1 in SW need to be secondarily considered.

## 5. Conclusions

In this study, we identified the FaAP2 family with 64 members from the *F. × ananassa* genome. The euAP2, euANT, and baselANT groups were identified by insertion motifs in two AP2 domains and phylogenetic analysis. Motif analysis proved that the R1 region-linker region–R2 region corresponded conserved region which is distinguished from variable pre-domain and post-domain regions. Synteny analysis indicated gene duplications and losses that explained the unequal AP2 loci of the sub-genomes. Six FaAP2 candidates—four WRI homologs and two AP2 homologs identified by expression profiles—could be regulators of red fruit color and/or special fruit aroma.

## Figures and Tables

**Figure 1 genes-12-00530-f001:**
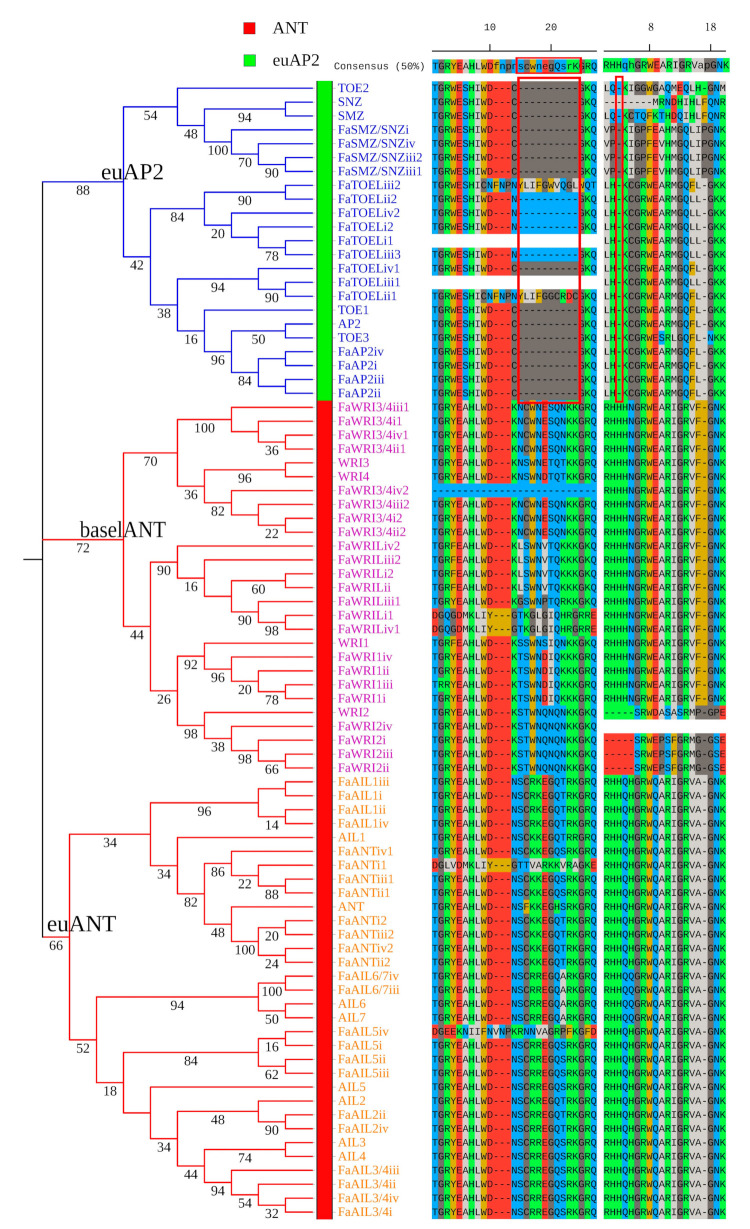
The maximum-likelihood (ML) phylogeny of APETALA2 (AP2) transcription factors of *Fragaria × ananassa* (FaAP2) and *Arabidopsis thaliana*. AINTEGUMENTA (ANT) and euAP2 lineages are indicated with green and red strips, respectively. The euAP2, baselANT, and euANT groups are distinguished by blue, pink, and yellow gene label colors, respectively. Numbers on branches are bootstrap values. Alignments of the insertion part of the AP2 domain repeat 1 and repeat 2 indicate the absent motifs of the euAP2 lineage underlined with red boxes.

**Figure 2 genes-12-00530-f002:**
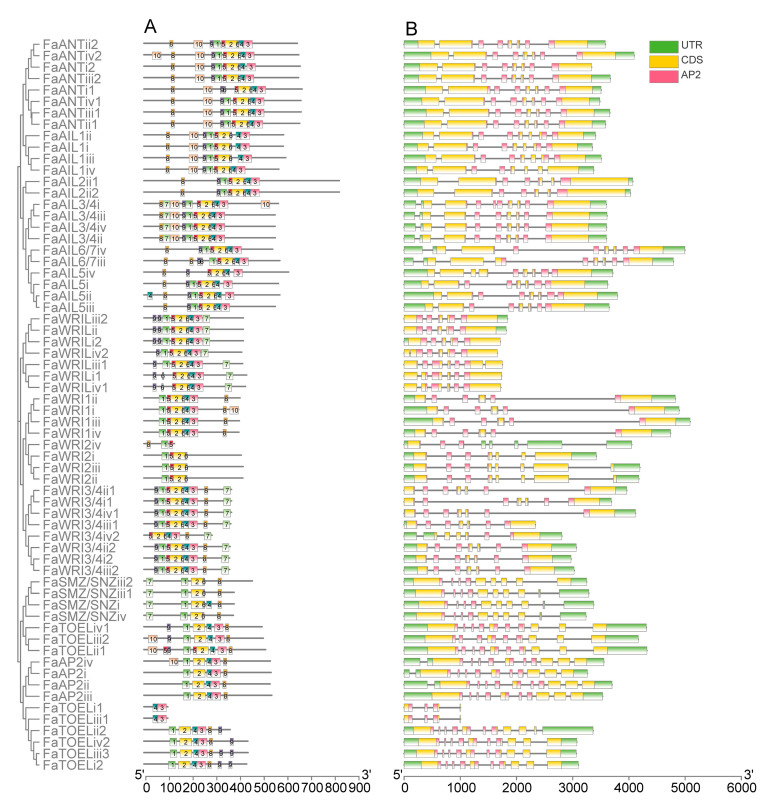
Motifs of AP2 proteins (**A**) and domain locations on the exon–intron structure of AP2 genes (**B**) of *Fragaria × ananassa*. ANT and euAP2 lineages are indicated with green and red strips, respectively. The euAP2, baselANT, and euANT groups are distinguished by blue, pink, and yellow labels, respectively. The numbers indicate 10 motifs, and the scale plate indicates the number of amino acid residues. The locations of the AP2 domains on exons (yellow) separated by introns (lines) are colored with red, and the scale plate indicates the number of nucleotides.

**Figure 3 genes-12-00530-f003:**
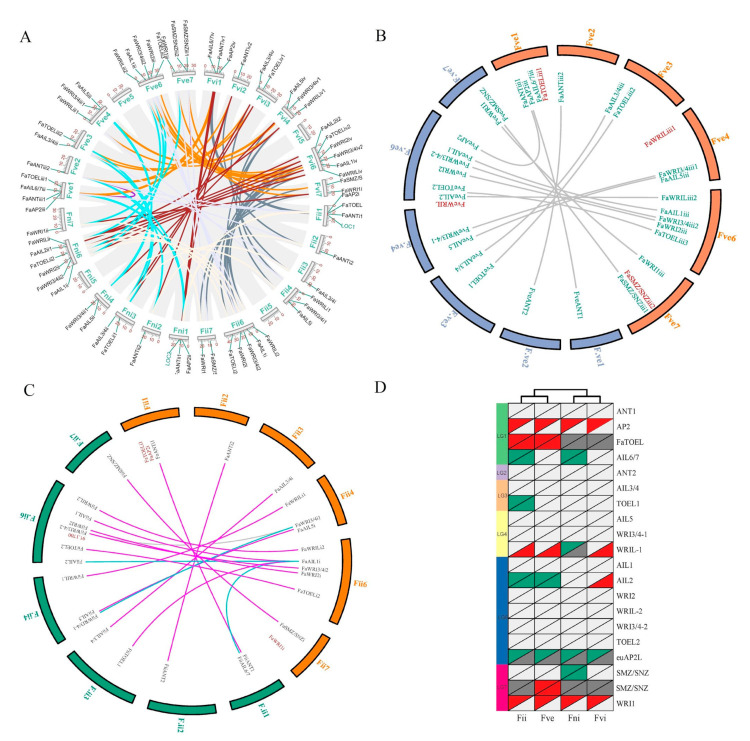
The intra-species and inter-species synteny analysis of the *Fragaria × ananassa* genome with sub-genome donors. (**A**) The intra-species synteny analysis. Syntenic regions are indicated by dark lines, and paralogous *AP2* pairs are highlighted. The 4 sub-genomes are named Fve, Fii, Fni, and Fvi according to sub-genome donors [2]. The 1–7 chromosomes of each sub-genome are arranged in order of the Fni, Fve, Fvi, and Fii sub-genomes. The locations of *AP2* genes are mapped by the scale plate value of the chromosomal length. Two green labeled homologous genes, *LOC1* and *LOC2*, were rejected as AP2 TFs due to a lack of the AP2 domain. (**B**) Synteny analysis of the Fve sub-genome of *F. × ananassa* with orange color between the *Fragaria vesca* genome whose chromosomes were named F.ve1–7. The orthologous *AP2* pairs are indicated by lines, and unpaired *AP2* TFs are red-labeled. (**C**) Synteny analysis of the Fii sub-genome of *F. × ananassa* with orange color between the *Fragaria iinumae* genome whose chromosomes were named F.ii1–7. The orthologous *AP2* pairs are indicated by lines, and unpaired *AP2* TFs are red-labeled. The pair lines supported/unsupported by phylogenetic analysis (Appendix A and S4) are highlighted with purplish red and blue colors, respectively. (**D**) Deductive gene duplication and loss. Gene loci are listed in 20 rows, and sub-genomes are listed in 4 columns. Red, blue, light gray, and dark gray triangles of the top left indicate gene duplication, gene loss, present shared locus, and absent shared locus compared with *F. iinumae AP2* loci, respectively. Low right triangles indicate an *F. vesca AP2* loci reference.

**Figure 4 genes-12-00530-f004:**
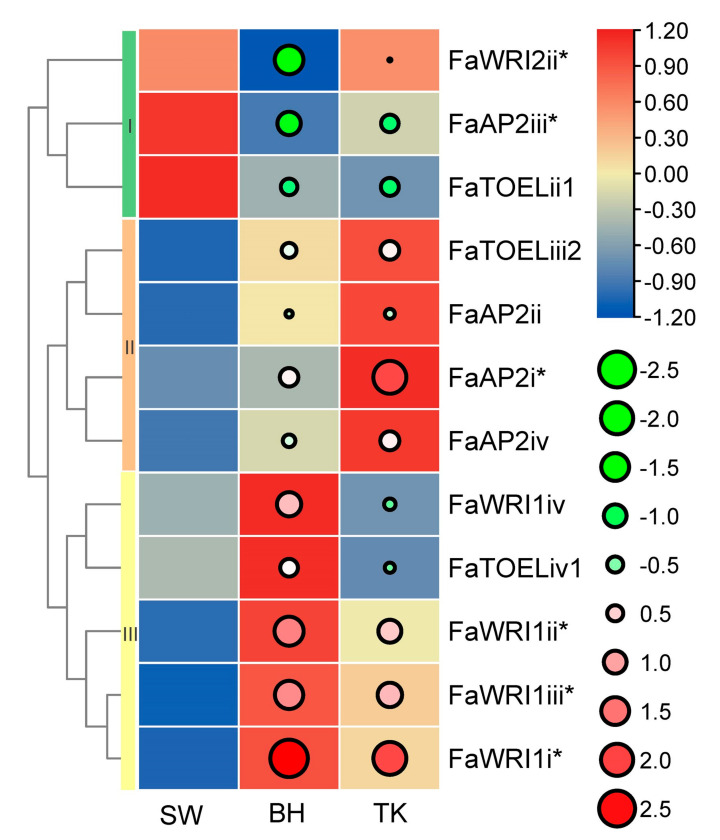
Expression profile of *AP2* genes in three strawberry cultivars. *AP2* genes are listed in rows, and the ‘Benihoppe,’ ‘Tokun,’ and ‘Snow White’ cultivars (SW, BH, and TK, respectively) are listed in columns. Normalized TPM (transcripts per kilobase of exon model per million mapped reads) are indicated by the gradient color of rectangles, and three row clusters are indicated by I, II, and III. Log2(fold-change) values using SW as a control sample are indicated by circle size and gradient color. * indicates significantly differentially expressed genes.

## Data Availability

Data is contained within the article or Appendix A. The data presented in this study are available.

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
