# Peer review of "Genome-Wide Screening of AP2 Transcription Factors Involving in Fruit Color and Aroma Regulation of Cultivated Strawberry"

_genes, 2021, doi:10.3390/genes12040530_

Round 1

Reviewer 1 Report

In general, I thought this manuscript represented a thorough comparative analysis of the AP2 gene family in cultivated strawberry. While it will likely be of interest to researchers associating agriculturally important traits (fruit flavor, etc) with specific regulators, there are several points that should be addressed. In general, the figures need to be better described, with better explanations added to the figure legends, and maybe a more friendly color palette used. Some of the figures were never mentioned in the text (Figure 2), and some information mentioned in the text was never described in a figure. I have elaborated on these and other comments below:

Comments associated with Figure 1: annotated the amino acid sequences along the top as R1 and R2. The amino acid sequences should be highlighted such that only those that agree with the consensus, or do not, are highlighted. As it is, the highlighting is not very useful or informative. Also, perhaps colors other than green/red could be used to distinguish the ANT/euAP2 groups?

Comments associated with Figure 2: inclusion of FaTOELi1, iii1, and FaWRI2iv? Are these pseudogenes? There is also no reference to Figure 2 in the text, although perhaps it should go around lines 149/150? This figure would be made easier for the non-expert by adding the euANT, baselANT, and euAP2 labels. Based on the depiction of TOELi1 and iii1, are these genes pseudogenes? What is the significance of the nomenclature for these genes?

Comments associated with Figure 3A: the grey and white lines are very hard to pick out. I would suggest using a color palette aid, such as ColorBrewer (if in R). What are your parameters for synteny? How many genes have to be part of a collinear block for the region to be considered syntenic? In 3B, maybe maybe it clearer that the orange/peach color is the F. vesca subgenome in ananassa (in the figure legend). Also, some of the lines do not match up with the genes they are supposed to link (for instance, FaANTiii1 and FaAP2iii are flipped). Figure 3C could use having the subgenome better demarcated as well. The analyses in 3A-3C heavily rely on an accurate assessment of synteny, that I cannot examine based on the methods/information provided. In 3D, what is LG supposed to indicate, both in the text and in the figure? The colors in the figure legend for 3D do not match those of the figure, making it difficult to determine what is occurring. And the LG nomenclature is very confusing. In addition, what if a gene was not present on the same “LG” but was present on a different chromosome, as demonstrated by 3A?

Comments associated with Figure 4: The methods here are woefully incomplete. How many sequencing replicates, where are the sequencing data deposited, how did they determine the differential expression?

In the discussion, line 234, do the authors ever demonstrate this in the results?

Line 252, grammatical errors and incomplete statement.

Line 128, "ATN" should be "ANT"

In the discussion, the authors include information about the red gene labels in 3B that should be included in the figure legend.

Author Response

REVIEWER1

In general, I thought this manuscript represented a thorough comparative analysis of the AP2 gene family in cultivated strawberry. While it will likely be of interest to researchers associating agriculturally important traits (fruit flavor, etc) with specific regulators, there are several points that should be addressed. In general, the figures need to be better described, with better explanations added to the figure legends, and maybe a more friendly color palette used. Some of the figures were never mentioned in the text (Figure 2), and some information mentioned in the text was never described in a figure. I have elaborated on these and other comments below:

Comments associated with Figure 1: annotated the amino acid sequences along the top as R1 and R2. The amino acid sequences should be highlighted such that only those that agree with the consensus, or do not, are highlighted. As it is, the highlighting is not very useful or informative. Also, perhaps colors other than green/red could be used to distinguish the ANT/euAP2 groups?

Answer: We have removed highlighting of sequences which do not agree of consensus alignment to make it more coordinating. The lineages and groups are easy for recognition with unchanged colors.

Comments associated with Figure 2: inclusion of FaTOELi1, iii1, and FaWRI2iv? Are these pseudogenes? There is also no reference to Figure 2 in the text, although perhaps it should go around lines 149/150? This figure would be made easier for the non-expert by adding the euANT, baselANT, and euAP2 labels. Based on the depiction of TOELi1 and iii1, are these genes pseudogenes? What is the significance of the nomenclature for these genes?

Answer: We have supplied the naming rules of gene in the text (genes location in 4 subgenomes were distinguished by i, ii, iii, iv). we have added the missing reference. Groups have been indicated with label colors.

Comments associated with Figure 3A: the grey and white lines are very hard to pick out. I would suggest using a color palette aid, such as ColorBrewer (if in R). What are your parameters for synteny? How many genes have to be part of a collinear block for the region to be considered syntenic? In 3B, maybe maybe it clearer that the orange/peach color is the F. vesca subgenome in ananassa (in the figure legend). Also, some of the lines do not match up with the genes they are supposed to link (for instance, FaANTiii1 and FaAP2iii are flipped). Figure 3C could use having the subgenome better demarcated as well. The analyses in 3A-3C heavily rely on an accurate assessment of synteny, that I cannot examine based on the methods/information provided. In 3D, what is LG supposed to indicate, both in the text and in the figure? The colors in the figure legend for 3D do not match those of the figure, making it difficult to determine what is occurring. And the LG nomenclature is very confusing. In addition, what if a gene was not present on the same “LG” but was present on a different chromosome, as demonstrated by 3A?

Answer: Exactly speaking, these blocks in 3A only seemed to be ‘syntenic regions’due to too many syntenic gene sets in these regions. Whether these blocks are real ‘syntenic genomic regions’ was not analyzed in our study but several studies have proved the broad and nice syntenic genomic regions in octaploid strawberry (e.g. Michael-A Hardigan et.al. Frontiers in Plant Science, 2020, 10.). Since this is a gene family study not a comparative genome study, we only focus on the collinear protein coding genes (not the genomic collinear regions). We did not set any special standard for identification of collinear block and the syntenic genes identification standard were definite in M&M. Anyway, we found few studies give a clear standard for identification of collinear block (only using of MCscan) and we looked forward reviewer to let us know ([email protected]).

We have revised the details of 3A-C (most lines were corrected because the orthologous pairs in 3B-C were identified by OrthoFinder). And we have changed LG (Linkage group) as Chr (Chromosome) as well as legend of 3D.

Comments associated with Figure 4: The methods here are woefully incomplete. How many sequencing replicates, where are the sequencing data deposited, how did they determine the differential expression?

Answer: We have added the details in the “2.5 Expression analysis in three cultivars”.

In the discussion, line 234, do the authors ever demonstrate this in the results?

Answer: We have supplied the miRNA response elements location and adjacent sequences comparing to prove it (Figure S5).

Line 252, grammatical errors and incomplete statement.

Answer: We have revised.

Line 128, "ATN" should be "ANT"

Answer: We have revised.

In the discussion, the authors include information about the red gene labels in 3B that should be included in the figure legend.

Answer: We have added in the legend.

Reviewer 2 Report

The present study has identified AP2 candidate TFs of F x ananassa genome. The manuscript is well-written and rich in references. The authors ended up with a list of candidate genes that can be potential regulators of red fruit color and/or fruit aroma. Besides, this interesting work, there are few things that should be modified to improve the manuscript.

M&M: please indicate the parameters used in each software and also the version, missing in line 78 and 80 for example. That information is missing in some of the following softwares.

Line 107: provide more information about the process. Explain how RNA was extracted, the methodology used for sequencing the transcriptome, and how was analyzed.

Line 132: mistyping subgenome

Line 138: provide some reference to Figure 2 in the text

Line 252: the title is incomplete

Figure 1: indicate to which lineages correspond the colors and also the outgroup

Figure 2: it could be easer for the readers that the tree has the same colors as in Figure 1

Figure S1: indicate the names at the end of the branches

Author Response

The present study has identified AP2 candidate TFs of F x ananassa genome. The manuscript is well-written and rich in references. The authors ended up with a list of candidate genes that can be potential regulators of red fruit color and/or fruit aroma. Besides, this interesting work, there are few things that should be modified to improve the manuscript.

M&M: please indicate the parameters used in each software and also the version, missing in line 78 and 80 for example. That information is missing in some of the following softwares.

Answer: We have added the parameters and versions.

Line 107: provide more information about the process. Explain how RNA was extracted, the methodology used for sequencing the transcriptome, and how was analyzed.

Answer: We have added the details.

Line 132: mistyping subgenome

Answer: We have revised.

Line 138: provide some reference to Figure 2 in the text

Answer: We have added.

Line 252: the title is incomplete

Answer: We have revised.

Figure 1: indicate to which lineages correspond the colors and also the outgroup

Answer: We have revised. Anyway, no outgroup was added since it is just a gene tree not a species tree.

Figure 2: it could be easer for the readers that the tree has the same colors as in Figure 1

Answer: We have revised the colors.

Figure S1: indicate the names at the end of the branches

Answer: We have tried to keep the gene names but it made the PDF too large to be browsed. We keep Figure S1 with unnamed branches and we would supply the NWK file (itol2 website would be a nice browsing tool) if anyone need the details of gene names.